

# Reference values for bone density and bone mineral content from 5 to 80 years old in a province of Chile

Marco Cossio-Bolanos[1,*], Rubén Vidal-Espinoza[2], Jose Fuentes-Lopez[3], Luis Felipe Castelli Correia de Campos[4], Cynthia Lee Andruske[5], Camilo Urra-Albornoz[6], Fernando Alvear Vasquez[7] and Rossana Gomez-Campos[8,*]

[1] Departamento de Ciencias de la Actividad Física, Universidad Católica del Maule, Maule, Talca, Chile
[2] Universidad Católica Silva Henriquez, Santiago, Chile
[3] Instituto de Investigación en Ciencias de la Educación (IICE), Escuela Profesional de Educación Física, Universidad Nacional del Altiplano de Puno, Puno, Perú
[4] Universidad del Bio Bio, Chillán, Chile
[5] Centro de Investigación CINEMAROS- SAC, Arequipa, Peru
[6] Escuela de Ciencias del Deporte y Actividad Física, Facultad de Salud, Universidad Santo Tomás, Talca, Chile
[7] Universidad de Valencia, Valencia, España
[8] Departamento de Diversidad e Inclusividad Educativa, Universidad Católica del Maule, Talca, Chile
* These authors contributed equally to this work.

Corresponding author
Rossana Gomez-Campos,
rossaunicamp@gmail.com

## ABSTRACT

**Background:** The assessment of bone health throughout the life cycle is essential to determine fracture risk. The objectives of the work were (a) compare bone mineral density and content with international references from the United States, (b) determine maximum bone mass, (c) propose references for bone health measurements from ages 5 to 80 years old.

**Methods:** Research was carried out on 5,416 subjects. Weight and height were measured. Body Mass Index (BMI) was calculated. The total body was scanned using dual energy X-ray absorptiometry (DXA). Information was extracted from the bone health measures (bone mineral density (BMD) and bone mineral content (BMC)) for both sexes, according to pediatric and adult software.

**Results and Discussion:** Differences were identified between the mean values of Chilean and American men for BMD (~0.03 to 0.11 $g/cm^2$) and BMC (~0.15 to 0.46 g). Chilean females showed average values for BMD similar to the US references (~ −0.01 to 0.02 $g/cm^2$). At the same time, they were relatively higher for BMC (~0.07 to 0.33 g). The cubic polynomial regression model reflected a relationship between BMD and BMC with chronological age in both sexes. For males, $R^2$ was higher ($R^2$ = 0.72 and 0.75) than for females ($R^2$ = 0.59 and 0.66). The estimate of maximum bone mass (MBM) for males emerged at 30 years old (1.45 ± 0.18 $g/cm^2$ of BMD and 3.57 ± 0.60 g of BMC) and for females at age 28 (1.22 ± 0.13 $g/cm^2$ of BMD and 2.57 ± 0.44 g of BMC). The LMS technique was used to generate smoothed percentiles for BMD and BMC by age and sex. Results showed that maximum bone mass occurred in females at age 28 and in males at 30. Reference values obtained from this research

may be used to evaluate bone health, diagnose bone fragility and osteoporosis in individuals and regional population groups.

# INTRODUCTION

During the past few years, preoccupation about bone health and fragility in children, adolescents, youth, and adults has generated great interest from researchers worldwide.

Childhood and adolescence are particularly important phases for maximizing the accumulation of bone tissue since the skeleton experiences rapid changes due to growth processes that shape and reshape the body in both males and females (*Baxter-Jones et al., 2011*). The bone mineral content (BMC) and bone mineral density (BMD) of the body increase as the size of the skeleton expands during growth (*Kralick & Zemel, 2020*). This process is characterized by rapid and significant longitudinal bone growth, expanded bone area, and accumulation of bone minerals (*Kindler, Lewis & Hamrick, 2015*). The development and accumulation of BMC and BMD during growth and biological maturation establishes the basis for lifelong skeletal health (*Guo et al., 2013*).

In fact, during the first two decades of life, the human skeleton grows in size as well as density. Furthermore, estimates indicate that more than half of the peak bone mass (PBM) is acquired during adolescence (*Bachrach, 2001*), reaching at the end of this stage more than 90% of the bone mass (*Bailey et al., 1999*; *Mccormack et al., 2017*). The remainder of the adult bone mass is acquired after the cessation of linear growth (*O'Flaherty, 2000*).

In general, the concept of maximum bone mass is defined more broadly similar to maximum bone strength. This is described in terms of mass, density, micro-structure, micro-reparatory mechanisms, and geometric properties that provide structural strength (*Weaver et al., 2016*).

The specific age at which PBM is reached is unclear, as the data are inconsistent. However, other studies conclude that PBM may reach beyond the third decade of life (*Baxter-Jones et al., 2011*; *Slosman et al., 1994*).

Regardless of the age at which it occurs, PBM remains relatively stable until about the fourth decade of life. Thereafter it begins to decline ostensibly in both sexes (*Weaver et al., 2016*; *Looker et al., 2012*), varying between individuals (*e.g.*, in women it is more accelerated than in men) and following different patterns according to age and sex, respectively (*Weaver et al., 2016*).

Successful acquisition of PBM depends on general health, illness, treatments, lifestyle factors during childhood and adulthood (*Rozenberg et al., 2020*). These include, for example, physical activity, dietary intake of calcium, smoking, alcohol consumption (*Waugh et al., 2009*), gender, and genetic composition (*Johnston & Slemenda, 1995*; *Herbert et al., 2019*) among others.

In general, alterations to skeletal health are of increasing concern throughout the life cycle. Therefore, examining BMD and BMC across the lifespan may be critical to understanding and mitigating fracture risk at older ages (*Hadji, Colli & Regidor, 2019*).
Consequently, as far as is known, specific reference values for examining skeletal health at all stages of life in regional, Chilean and South American populations are scarce. Except for the references proposed for the United States (*Looker et al., 2012*) and for the Mexican population (*Tamayo et al., 2009*), however, due to the heterogeneity between regions and populations worldwide, it is necessary for each country to have its own references.

In fact, Chile is one of the most prosperous countries in Latin America, and is experiencing a demographic transition, characterized by an expanding aging population and a declining underpopulation.

In addition, between 2006 and 2017, osteoporosis fractures increased more than 40% (*Quevedo et al., 2020*). Fractures create serious public health problems in Chile. Therefore, studying the bone health of a sample population from childhood to old age is important and relevant for public health. Such information could help describe skeletal health patterns and identify PBM in both sexes. It may also help to develop preventive health measures and propose public health policies for Chile and neighboring South American countries. Therefore, the objectives of this study were: (a) to compare bone density and bone mineral content references with American and international studies; (b) to determine maximum bone mass; and (c) to propose provincial bone health references from 5 to 80 years for the population of a province in Chile.

## METHODS

### Type of study and sample

The population of the Maule region (Chile) consisted of 41,200 subjects between 5.0 and 80.0 years of age. Probabilistic (random) sampling was used to calculate the sample size, which resulted in 5,416 subjects (2,847 (6.9%) males and 2,569 (6.3%) females) with a 95% CI. Children, adolescents, youth, adults, and older adults were recruited as volunteers for the study.

Volunteers were recruited from public schools, universities (public and private), social and business programs sponsored by the Municipality of the Maule region. The Maule is the seventh region in Chile with Talca as its capital city. The Maule is composed of four provinces: Cauquenes, Curicó, Linares, and Talca. As depicted in Fig. 1 it is located 250 km south of Santiago, the capital of Chile (Fig. 1). The region is 102 m above sea level.

According to the *United Nations Development Programme (2019)*, the Human Development Index (HDI) of Chile for 2018 was 0.847. Life expectancy is 80 years. In Chile, the educational system is divided into four stages (kindergarten, elementary, middle and higher education), being compulsory on average between 5 and 17 years of age. In the Maule Region, HDI was 0.872.

Figure 2 shows the description diagram of participating subjects. Children and adolescent students, university students, and adults completing the anthropometric assessments and the DXA scans were included in the study. Subjects excluded from the research were those of another nationality other than Chilean (nationality was verified on each student's enrollment form), those with a physical disability, and those with any type of metal prosthesis and/or implant that could interfere with the DXA scan. Subjects who reported having suffered from metabolic bone disorders, bone fractures in the last 6
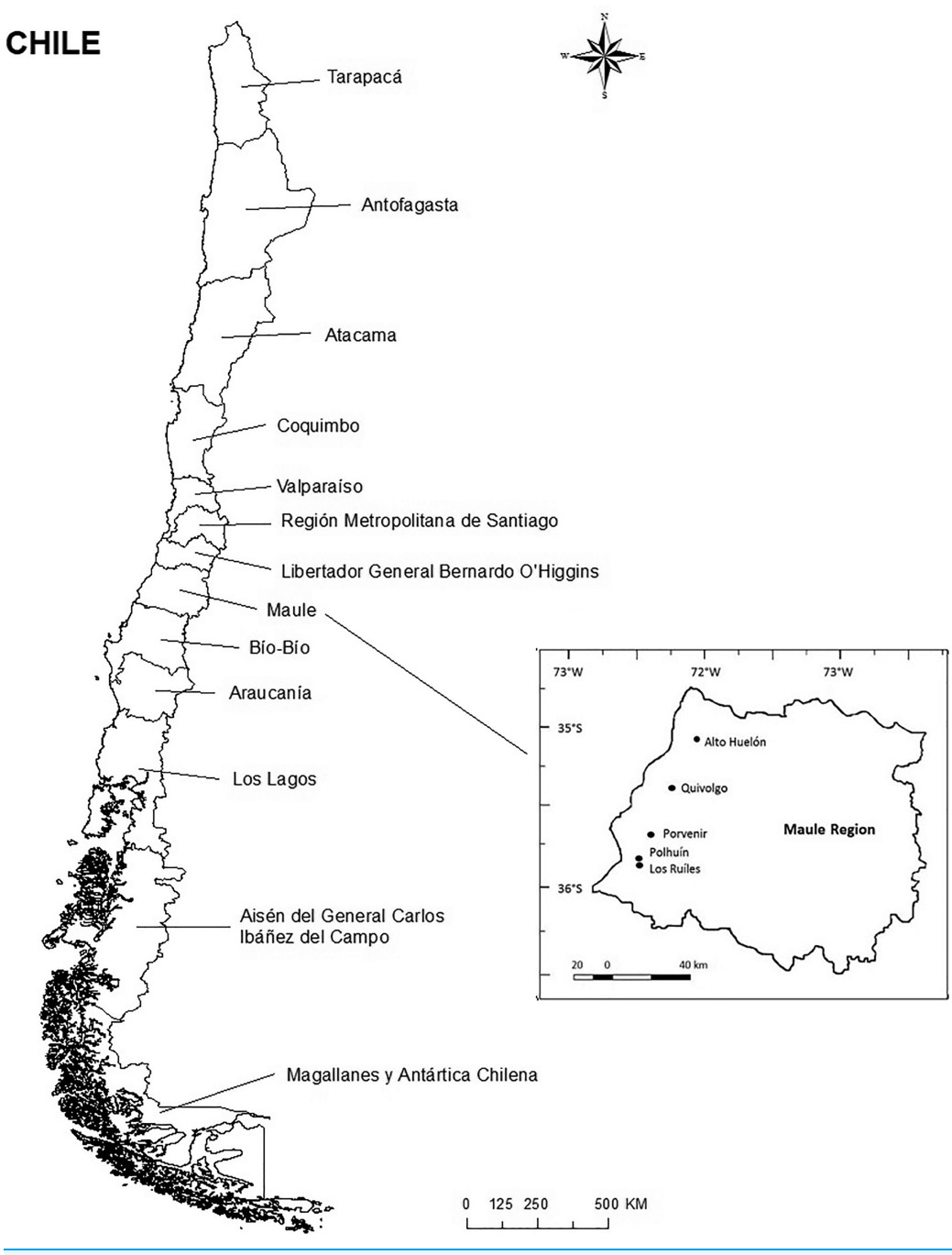

Figure 1 Location of the study province in Chile.

months (four children and six adults) and habitual smokers (22 males and 18 females) were also excluded. All volunteers (minors and adults) signed the informed consent form authorizing the anthropometric and DXA scan evaluations. In the case of minors, parents and/or guardians signed the consent for their children from 5 to 17.9 years of age. In addition, these same children signed the assent from ages 8 to 17.9 years. However, children aged 5, 6 and 7 years were assisted by their parents to authorize the assent

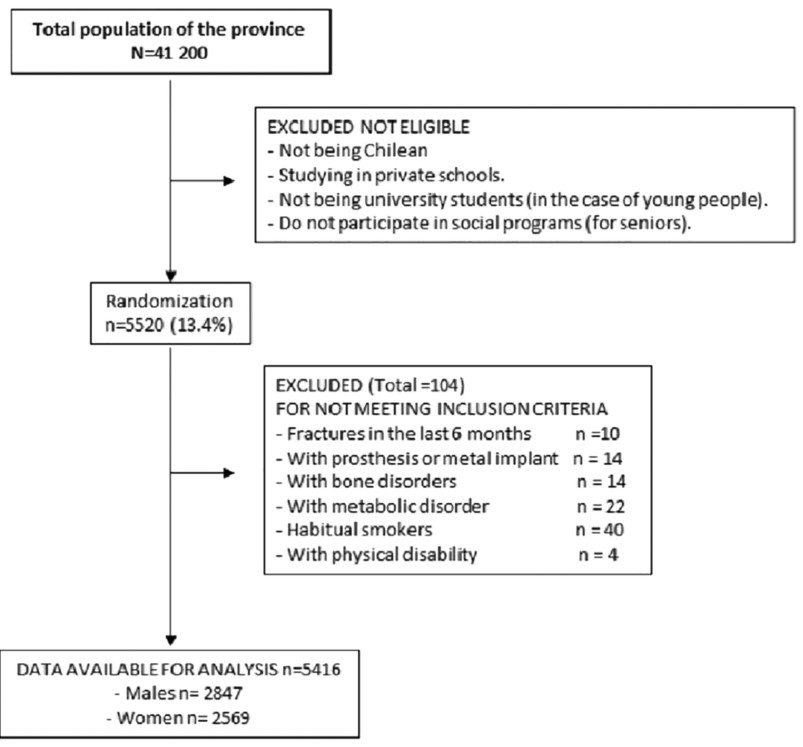

**Figure 2** Diagram of description of participating subjects.

(by marking on the form). The research was carried out according to the Declaration of Helsinki for Human Subjects and was approved by the University Ethics Committee (UA-238-2014).

## PROCEDURES AND MEASURES

Data for age, sex, school, university, place of work, and social program were collected for each individual and recorded in a separate file created by the researchers. All evaluations were carried out in a closed laboratory (20 °C to 24 °C) at the University. The subjects were evaluated during 2015, 2016, 2017, and 2018 from 8:30 a.m. to 13:00 p.m. and 15:00 to 18:00 p.m., Monday to Friday, from April to June and August to November each year of the study.

The anthropometric assessments for weight and height were collected based on the standard protocol developed by *Ross & Marfell-Jones (1991)*. Assessments were conducted with the volunteer wearing the least amount of clothing possible (shorts, shirt, and no shoes). For body weight (kg), an electric scale (Tanita, United Kingdom) with a scale of 0 to 150 kg and an accuracy of 100 g was used. Standing height was measured according to the Frankfurt Plane with a portable stadiometer (Seca Gmbh & Co. KG, Hamburg, Germany) with a precision of 0.1 mm. Body Mass Index (BMI) was calculated with the formula: BMI = weight (kg)/height$^2$ (m).

Scanning was performed with the dual energy X-ray absorptiometry (DXA) (Lunar Prodigy; General Electric, Fairfield, CT, USA). The procedure consisted of scanning the whole body. For the analysis of the measurements, the head was excluded in all subjects. The examinations were carried out with a single densitometer using software specifically for adults and pediatric populations (<18 years). Values were extracted for the bone mineral content (BMC) and bone mineral density (BMD).

The procedure consisted of the subjects lying down on the exploration platform in a supine position with their arms and legs extended (in pronation). The ankles are held together with Velcro belt to ensure standard positioning. The subjects for this study were warned that any type of metal on the body could impede the scan. Two trained technicians with significant experience were in charge of calibrating the equipment every day according to the manufacturer's instructions.

To guarantee quality control for the variables collected, 10% of the total sample was evaluated twice (285 males and 257 females). The technique of test-re-test for anthropometry and DXA scan was carried out. The technical error of measurement (TEM) for the anthropometric variables showed values between 0.8% and 1.2%, and DXA values for both BMD and BMC ranged from 0.7% to 1.1%.

Comparisons of the BMD and BMC were performed with the references from the Center for National Health Statistics for the United States, NHANES (which includes all races and ethnicities and non-Hispanic whites) (*Looker et al., 2012*). This reference presented age groups: 8–11y, 12–15y, 16–19y, 20–30y, 30–40y, 40–50y, 50–60y, 60–70y, and 70–80y for both sexes.

## Statistics

The normal distribution of the data was determined by using the Kolmogorov–Smirnov's test. Descriptive statistical analysis of the arithmetic mean and the standard deviation were performed. Differences between sexes were determined with the test for independent student samples. Comparisons with the references from the United States were calculated using the fraction 100 log (reference percentile/calculated percentile). Different regression analysis models were used, with the third degree cubic polynomial model as the most adequate for both sexes: BMD and BMC = a + b1(age) + b2(age)$^2$ + b3(age)$^3$, where a is the intercept, b1, b2, and b3 are regression parameters, estimated from the data. All analyses were performed separately for both men and women. The best model was selected based on $R^2$ and statistical significance.

Based on the parameters of the polynomial regression model, the maximum means for BMD and BMD and ±SD were determined. For all of the calculations performed, $P < 0.05$ was adopted for significance. Percentile curves (P3, P5, P10, P15, P25, P50, P75, P85, P90, P95, and P97) were created for BMD and BMC for each age group and sex based on the LMS method (*Cole et al., 2000*). LMS Chart Maker version 2.3 software (*Pan & Cole, 2006*) was used. Final percentile curves were smoothed to create three specific curves for age: L (Lambda; asymmetry), M (Mu; median), and S (Sigma; coefficient of variation). The significance adopted was 0.001. Calculations were performed on Excel sheets and SPSS 16.0.

## RESULTS

The variables for weight, height, BMI, BMD, and BMC are illustrated in Table 1. There were no significant differences between both sexes in body weight and BMD from 5 to 14 years of age. However, from age 15 to 80 years old, males were heavier and had greater BMD when compare to the females. For height, no differences emerged from ages 5 to 12. From 13 to 80 years old, females had a lower CMO compared to men. No significant differences resulted in BMI in all ages. Comparison of the mean values between BMD and BMC of the group in the Maule Region (Chile) and the references from the United States are shown in Fig. 3. The mean values for BMD (~0.03 to 0.11 $g/cm^2$) and BMC (~0.15 to 0.46 g) of the males were relatively greater than the reference values for the United States. In the case of women, the mean BMD values compared to the American references were relatively similar (~−0.01 to 0.02 $g/cm^2$). However, in terms of BMC, Chilean women had higher BMC (~0.07 to 0.33 g).

The polynomial cubic regression model indicated a relationship between the BMD and BMC with chronological age in both sexes (Fig. 4). The $R^2$ for males was higher ($R^2 = 0.72$ and 0.75) than that of the females ($R^2 = 0.59$ and 0.66). The estimation of the PBM at age 30 for males emerged as ($1.45 \pm 0.18$ $g/cm^2$ of BMD and $3.57 \pm 0.60$ g of BMC) and females at age 28 ($1.22 \pm 0.13$ $g/cm^2$ of BMD and $2.57 \pm 0.44$ g of BMC). In general, the polynomial regression model showed an accelerated growth phase for bone mass in both sexes (age 5 until 28–30 years old) followed by a plateau without significant gains or losses until the age of 40 years. After this age, they undergo declines that continue until advanced ages (in males, BMD decreased by 17.48% and BMC by 24.30%, and in females, BMD decreased by 20.50% and BMC by 24.50%).

The percentiles for BMD and BMC by age and sex are presented in Tables 2 and 3 (P3, P5, P10, P15, P25, P50, P75, P85, P90, P95, and P97). The values for BMD in both sexes were similar from age 5 until age 13. After which, values have a higher median compared to females. This difference persists until old age, however, males have higher CMO values compared to females at all ages, and the greatest difference manifests itself at age 10 and persists until age 80.

## DISCUSSION

The results of the study showed that Chilean men from Maule have marginally higher BMC and BMD values compared to the U.S. population, according to NHANES data. While Chilean women showed similar values *versus* their U.S. counterparts (as calculated by the fraction 100 logarithm to compare regional percentiles).

In fact, from age 6 to 20 years old, the bone health (BMD) values in this study were similar to those reflected in children and adolescents form the less developed countries (*van der Sluis et al., 2002*). In addition, these average values were identical during the growth phase and late adulthood (until 80 years of age) with the Mexican population (*Tamayo et al., 2009*) and similar to the population of Singapore (*Chen et al., 2020*), whose skeletal health indicators have been tested by DEXA. Also, the results illustrated that the BMD values of the regional population of the Maule (Chile), especially during childhood

**Table 1 Anthropometric characteristics, mean values, and ±SD for bone mineral density and content for both sexes.**

| Age (y) | | Weight (kg) | | p | Height (cm) | | p | BMI (kg/m²) | | p | BMD (g/m²) | | p | BMC (g) | | p |
|---|---|---|---|---|---|---|---|---|---|---|---|---|---|---|---|---|
| | n | X | SD | | X | SD | | X | SD | | X | SD | | X | SD | |
| **Males** | | | | | | | | | | | | | | | | |
| 5.0–5.9 | 91 | 21.99 | 3.23 | 0.5573 | 113.85 | 5.77 | 0.8767 | 16.93 | 1.88 | 0.4885 | 0.72 | 0.11 | 0.1588 | 0.86 | 0.38 | 0.2005 |
| 6.0–6.9 | 105 | 26.02 | 6.24 | 0.913 | 120.34 | 5.76 | 0.4058 | 17.86 | 3.50 | 0.7796 | 0.75 | 0.06 | >0.999 | 0.93 | 0.18 | 0.7366 |
| 7.0–7.9 | 103 | 30.13 | 7.17 | 0.3853 | 126.64 | 6.41 | 0.6466 | 18.63 | 3.37 | 0.4211 | 0.79 | 0.07 | 0.0605 | 1.08 | 0.23 | 0.1282 |
| 8.0–8.9 | 116 | 31.81 | 6.68 | 0.6918 | 130.36 | 5.10 | 0.6917 | 18.59 | 3.03 | 0.4764 | 0.81 | 0.09 | 0.0642 | 1.17 | 0.32 | 0.0914 |
| 9.0–9.9 | 126 | 35.96 | 8.07 | 0.3158 | 135.35 | 6.58 | 0.0797 | 19.54 | 3.76 | 0.6559 | 0.84 | 0.08 | 0.3806 | 1.28 | 0.28 | 0.5791 |
| 10.0–10.9 | 120 | 41.50 | 9.17 | 0.6294 | 142.09 | 6.85 | 0.0609 | 20.38 | 3.26 | 0.6493 | 0.88 | 0.09 | 0.0729 | 1.44 | 0.32 | 0.2626 |
| 11.0–11.9 | 136 | 46.31 | 11.83 | 0.5541 | 147.44 | 7.71 | 0.0719 | 21.13 | 4.01 | 0.3875 | 0.92 | 0.08 | >0.999 | 1.60 | 0.31 | 0.792 |
| 12.0–12.9 | 143 | 49.12 | 10.96 | 0.2628 | 153.69 | 7.81 | 0.0056 | 20.73 | 4.05 | 0.0012 | 0.95 | 0.08 | 0.063 | 1.79 | 0.29 | 0.0671 |
| 13.0–13.9 | 190 | 54.31 | 10.94 | 0.253 | 160.33* | 8.57 | <0.0001 | 21.01 | 3.33 | 0.0002 | 1.02 | 0.12 | 0.1508 | 2.05* | 0.42 | 0.078 |
| 14.0–14.9 | 197 | 60.09 | 11.91 | 0.4221 | 165.90* | 6.88 | <0.0001 | 21.77 | 3.91 | 0.0001 | 1.07 | 0.11 | >0.999 | 2.29* | 0.40 | <0.0001 |
| 15.0–15.9 | 222 | 64.61* | 9.05 | 0.003 | 170.20* | 6.78 | <0.0001 | 22.30 | 2.94 | 0.0012 | 1.16* | 0.11 | <0.0001 | 2.62* | 0.37 | <0.0001 |
| 16.0–16.9 | 247 | 70.84* | 12.98 | <0.001 | 171.67* | 7.43 | <0.0001 | 24.00 | 3.93 | 0.4439 | 1.22* | 0.13 | <0.0001 | 2.81* | 0.41 | <0.0001 |
| 17.0–17.9 | 292 | 71.38* | 12.36 | <0.001 | 171.63* | 6.11 | <0.0001 | 24.19 | 3.72 | 0.0294 | 1.23* | 0.11 | <0.0001 | 2.82* | 0.38 | <0.0001 |
| 18.0–18.9 | 152 | 71.82* | 11.07 | <0.001 | 172.14* | 6.90 | <0.0001 | 24.25 | 3.62 | 0.5143 | 1.25* | 0.10 | <0.0001 | 2.92* | 0.38 | <0.0001 |
| 19.0–19.9 | 91 | 71.87* | 9.57 | <0.001 | 172.39* | 6.09 | <0.0001 | 24.15 | 2.65 | 0.0124 | 1.26* | 0.12 | <0.0001 | 2.95* | 0.43 | <0.0001 |
| 20.0–30.0 | 322 | 77.31* | 12.28 | <0.001 | 173.12* | 6.66 | <0.0001 | 25.76 | 3.62 | 0.0100 | 1.29* | 0.12 | <0.0001 | 3.07* | 0.44 | <0.0001 |
| 30.0–40.0 | 37 | 82.86* | 12.78 | <0.001 | 175.30* | 9.37 | <0.0001 | 26.93 | 3.35 | 0.6590 | 1.29* | 0.18 | <0.0001 | 3.13* | 0.68 | <0.0001 |
| 40.0–50.0 | 32 | 84.28* | 15.36 | <0.001 | 169.99* | 5.49 | <0.0001 | 29.10 | 4.64 | 0.2827 | 1.23* | 0.11 | <0.0001 | 2.91* | 0.34 | <0.0001 |
| 50.0–60.0 | 43 | 86.85* | 11.61 | <0.001 | 169.55* | 6.81 | <0.0001 | 30.25 | 3.91 | 0.1075 | 1.25* | 0.12 | <0.0001 | 2.89* | 0.39 | <0.0001 |
| 60.0–70.0 | 43 | 81.52* | 12.98 | <0.001 | 167.87* | 6.63 | <0.0001 | 28.85 | 3.75 | 0.1548 | 1.20* | 0.13 | <0.0001 | 2.83* | 0.44 | <0.0001 |
| 70.0–80.0 | 39 | 80.40* | 12.67 | <0.001 | 166.11* | 6.28 | <0.0001 | 29.07 | 3.76 | 0.5236 | 1.18* | 0.14 | <0.0001 | 2.75* | 0.38 | <0.0001 |
| **Females** | | | | | | | | | | | | | | | | |
| 5.0–5.9 | 83 | 22.28 | 3.27 | | 113.98 | 5.21 | | 17.14 | 2.11 | | 0.70 | 0.07 | | 0.80 | 0.20 | |
| 6.0–6.9 | 77 | 25.92 | 5.88 | | 119.58 | 6.49 | | 18.00 | 3.08 | | 0.75 | 0.08 | | 0.92 | 0.22 | |
| 7.0–7.9 | 88 | 29.30 | 5.79 | | 126.24 | 5.48 | | 18.27 | 2.69 | | 0.77 | 0.07 | | 1.03 | 0.22 | |
| 8.0–8.9 | 88 | 32.19 | 6.89 | | 130.06 | 5.65 | | 18.90 | 3.13 | | 0.78 | 0.08 | | 1.10 | 0.25 | |
| 9.0–9.9 | 98 | 37.06 | 8.19 | | 137.72 | 6.96 | | 19.38 | 3.12 | | 0.83 | 0.09 | | 1.26 | 0.25 | |
| 10.0–10.9 | 145 | 42.05 | 9.27 | | 143.84 | 7.53 | | 20.19 | 3.48 | | 0.86 | 0.09 | | 1.40 | 0.26 | |
| 11.0–11.9 | 107 | 47.14 | 9.43 | | 150.69 | 7.10 | | 20.70 | 3.62 | | 0.92 | 0.09 | | 1.61 | 0.27 | |
| 12.0–12.9 | 142 | 54.44 | 10.97 | | 156.02 | 6.02 | | 22.29 | 4.01 | | 1.00 | 0.10 | | 1.88 | 0.27 | |
| 13.0–13.9 | 96 | 55.90 | 11.37 | | 156.84 | 7.34 | | 22.68 | 4.06 | | 1.04 | 0.09 | | 1.99 | 0.30 | |
| 14.0–14.9 | 106 | 58.95 | 11.51 | | 157.77 | 8.29 | | 23.56 | 3.74 | | 1.07 | 0.11 | | 2.06 | 0.33 | |
| 15.0–15.9 | 83 | 60.07 | 10.72 | | 159.20 | 4.35 | | 23.70 | 4.17 | | 1.07 | 0.09 | | 2.08 | 0.26 | |
| 16.0–16.9 | 121 | 62.75 | 11.90 | | 159.16 | 5.42 | | 24.75 | 4.47 | | 1.09 | 0.08 | | 2.14 | 0.25 | |
| 17.0–17.9 | 125 | 63.14 | 13.30 | | 158.27 | 4.96 | | 25.14 | 4.78 | | 1.10 | 0.08 | | 2.14 | 0.22 | |
| 18.0–18.9 | 60 | 59.51 | 9.71 | | 157.81 | 5.88 | | 23.89 | 3.60 | | 1.10 | 0.09 | | 2.15 | 0.26 | |
| 19.0–19.9 | 55 | 66.61 | 11.97 | | 161.30 | 6.16 | | 25.53 | 3.93 | | 1.15 | 0.10 | | 2.28 | 0.31 | |
| 20.0–30.0 | 240 | 64.26 | 12.52 | | 160.60 | 6.05 | | 24.88 | 4.44 | | 1.14 | 0.09 | | 2.29 | 0.30 | |
| 30.0–40.0 | 71 | 69.49 | 10.60 | | 159.83 | 8.04 | | 27.30 | 4.47 | | 1.14 | 0.09 | | 2.33 | 0.27 | |

| Age (y) | | | p | Height (cm) | | p | BMI (kg/m²) | | p | BMD (g/m²) | | p | BMC (g) | | p |
|---|---|---|---|---|---|---|---|---|---|---|---|---|---|---|---|
| | n | X | SD | | X | SD | | X | SD | | X | SD | | X | SD | |

| Age (y) | n | Weight (kg) X | SD | p | Height (cm) X | SD | p | BMI (kg/m²) X | SD | p | BMD (g/m²) X | SD | p | BMC (g) X | SD | p |
|---|---|---|---|---|---|---|---|---|---|---|---|---|---|---|---|---|
| 40.0–50.0 | 110 | 69.96 | 11.23 | | 157.58 | 6.69 | | 28.17 | 4.19 | | 1.12 | 0.10 | | 2.27 | 0.27 | |
| 50.0–60.0 | 163 | 70.43 | 12.37 | | 156.22 | 5.63 | | 28.90 | 5.09 | | 1.08 | 0.11 | | 2.16 | 0.27 | |
| 60.0–70.0 | 293 | 69.93 | 11.32 | | 152.97 | 6.67 | | 29.94 | 4.80 | | 1.03 | 0.11 | | 2.05 | 0.28 | |
| 70.0–80.0 | 218 | 67.65 | 12.29 | | 151.11 | 6.26 | | 29.57 | 4.62 | | 0.97 | 0.13 | | 1.94 | 0.33 | |

**Notes:**
* Significant difference in relation to women in the same age group.
X, mean; SD, standard deviation; BMD, bone mineral density; BMC, bone mineral content; BMI, Body Mass Index.

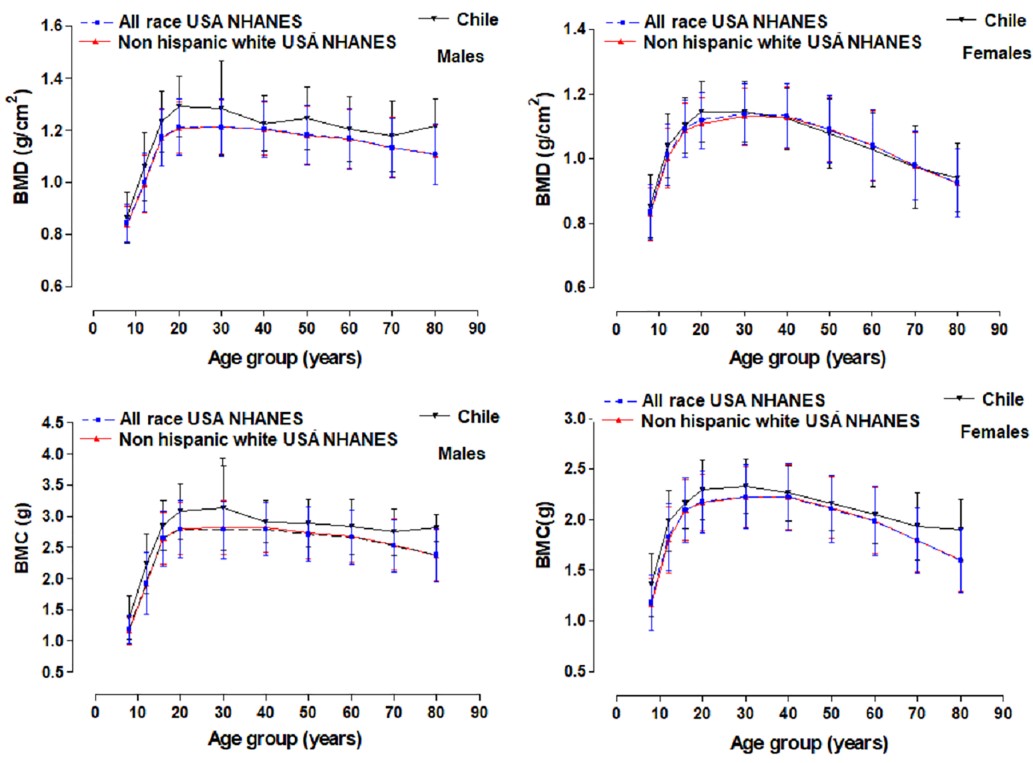

**Figure 3 Graphic comparison of mean values for BMD and BMC for the United States references (NHANES) with the group studied in a province of Chile.**

and adolescence, presented average values much higher than those related to China (*Hao et al., 2007*) and Korea (*Yi et al., 2014*).

These findings are consistent with other research that has reported population differences in BMD and BMC across the life span (*Looker et al., 2012*; *Nakavachara et al., 2014*; *Chen et al., 2020*; *Hoiberg et al., 2007*; *Noon et al., 2010*; *Ho-Pham et al., 2011*). These differences in skeletal health values in various geographic regions, populations and ethnicities may be due to several factors such as diet, nutritional status, physical exercise, vitamin D intake, and daily activities. Even other important factors are annual sun

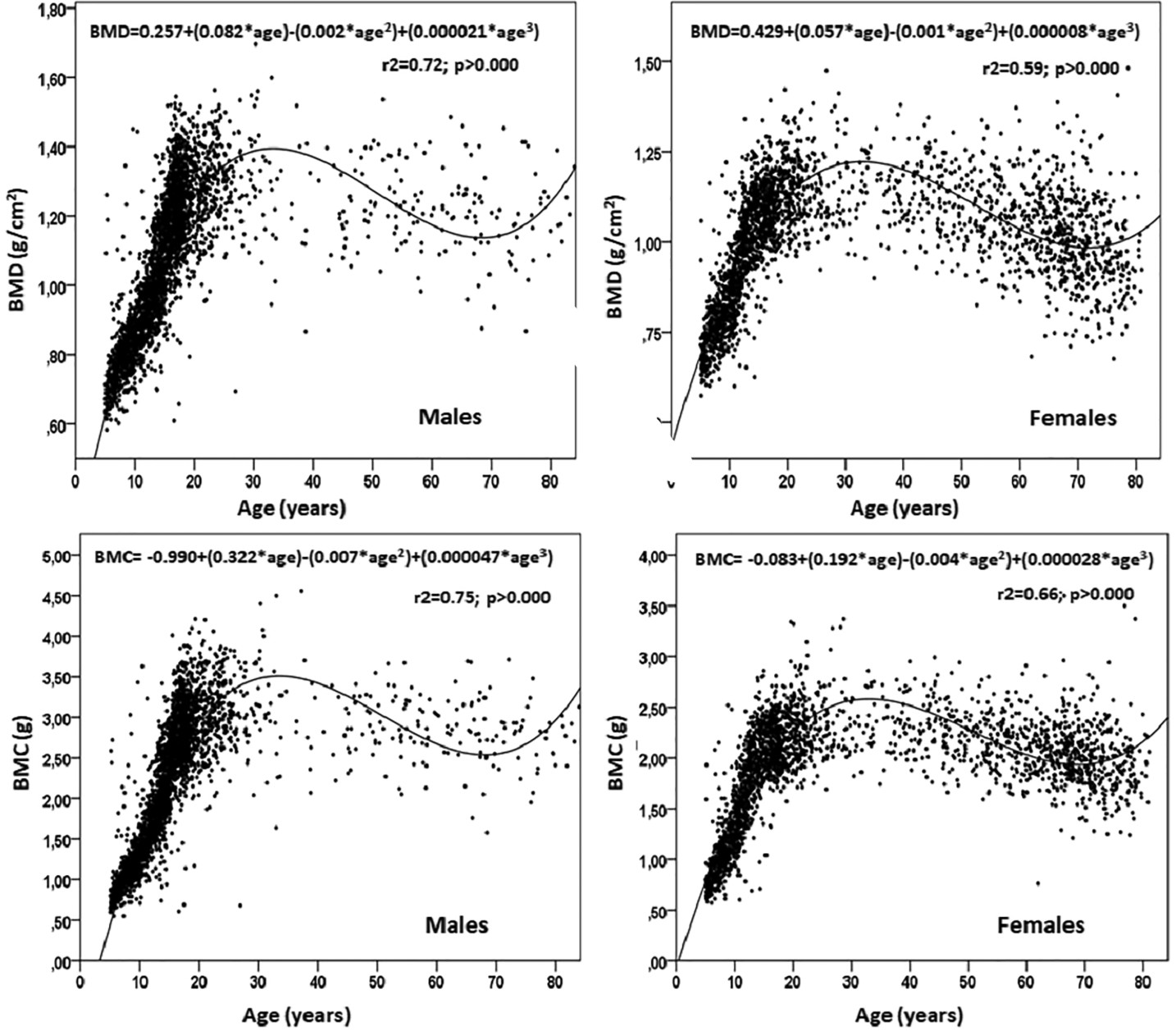

**Figure 4 Results of the quadratic relationship between chronological age and bone mineral density and content of males and females from a province of Chile.**

exposure, lifestyle, demographic characteristics, education levels, and general health (*Kralick & Zemel, 2020*; *Nakavachara et al., 2014*; *Heidari et al., 2016*).

Regarding the relationship between BMD and BMC with chronological age, this study showed that the cubic polynomial regression model had greater explanatory power for both sexes. This relationship between age and mineral measurements were closer in males (72–75%) compared to females (59–66%). These relationships are consistent with studies conducted in populations with children, youth, and adults of both sexes (*Chen et al., 2020*; *Ho-Pham et al., 2011*; *Heidari et al., 2016*), respectively.

**Table 2 Total body bone mineral density (g/m²) from 5.0 to 80 years old of subjects from a province of Chile.**

| Age | L | M | S | P3 | P5 | P10 | P15 | P25 | P50 | P75 | P85 | P90 | P95 | P97 |
|---|---|---|---|---|---|---|---|---|---|---|---|---|---|---|
| **Males** | | | | | | | | | | | | | | |
| 5.0–5.9 | −2.78 | 0.70 | 0.08 | 0.62 | 0.63 | 0.64 | 0.65 | 0.67 | 0.70 | 0.75 | 0.78 | 0.80 | 0.84 | 0.87 |
| 6.0–6.9 | −2.35 | 0.74 | 0.09 | 0.64 | 0.65 | 0.67 | 0.68 | 0.70 | 0.74 | 0.78 | 0.81 | 0.84 | 0.87 | 0.90 |
| 7.0–7.9 | −1.91 | 0.77 | 0.09 | 0.67 | 0.68 | 0.70 | 0.71 | 0.73 | 0.77 | 0.82 | 0.85 | 0.87 | 0.91 | 0.94 |
| 8.0–8.9 | −1.48 | 0.80 | 0.09 | 0.69 | 0.70 | 0.72 | 0.74 | 0.76 | 0.80 | 0.86 | 0.89 | 0.91 | 0.95 | 0.98 |
| 9.0–9.9 | −1.05 | 0.84 | 0.09 | 0.72 | 0.73 | 0.75 | 0.77 | 0.79 | 0.84 | 0.90 | 0.93 | 0.95 | 0.99 | 1.01 |
| 10.0–10.9 | −0.62 | 0.88 | 0.09 | 0.75 | 0.76 | 0.78 | 0.80 | 0.83 | 0.88 | 0.94 | 0.97 | 1.00 | 1.03 | 1.06 |
| 11.0–11.9 | −0.19 | 0.93 | 0.09 | 0.78 | 0.79 | 0.82 | 0.84 | 0.87 | 0.93 | 0.99 | 1.02 | 1.05 | 1.08 | 1.11 |
| 12.0–12.9 | 0.22 | 0.98 | 0.10 | 0.81 | 0.83 | 0.86 | 0.88 | 0.91 | 0.98 | 1.04 | 1.08 | 1.10 | 1.14 | 1.16 |
| 13.0–13.9 | 0.60 | 1.03 | 0.10 | 0.85 | 0.87 | 0.91 | 0.93 | 0.96 | 1.03 | 1.10 | 1.14 | 1.16 | 1.20 | 1.23 |
| 14.0–14.9 | 0.93 | 1.09 | 0.10 | 0.89 | 0.91 | 0.95 | 0.98 | 1.02 | 1.09 | 1.16 | 1.20 | 1.22 | 1.26 | 1.29 |
| 15.0–15.9 | 1.22 | 1.14 | 0.10 | 0.93 | 0.95 | 1.00 | 1.02 | 1.07 | 1.14 | 1.22 | 1.26 | 1.28 | 1.32 | 1.35 |
| 16.0–16.9 | 1.47 | 1.19 | 0.10 | 0.96 | 0.99 | 1.04 | 1.07 | 1.11 | 1.19 | 1.27 | 1.31 | 1.33 | 1.37 | 1.40 |
| 17.0–17.9 | 1.68 | 1.23 | 0.10 | 0.98 | 1.02 | 1.07 | 1.10 | 1.14 | 1.23 | 1.31 | 1.35 | 1.37 | 1.41 | 1.44 |
| 18.0–18.9 | 1.84 | 1.25 | 0.10 | 1.00 | 1.04 | 1.09 | 1.12 | 1.17 | 1.25 | 1.33 | 1.38 | 1.40 | 1.44 | 1.47 |
| 19.0–19.9 | 1.98 | 1.27 | 0.10 | 1.01 | 1.05 | 1.10 | 1.14 | 1.19 | 1.27 | 1.35 | 1.39 | 1.42 | 1.46 | 1.49 |
| 20.0–30.0 | 2.09 | 1.28 | 0.10 | 1.02 | 1.06 | 1.11 | 1.15 | 1.20 | 1.28 | 1.36 | 1.40 | 1.43 | 1.47 | 1.50 |
| 30.0–40.0 | 2.18 | 1.28 | 0.10 | 1.02 | 1.06 | 1.11 | 1.15 | 1.20 | 1.28 | 1.37 | 1.41 | 1.43 | 1.47 | 1.50 |
| 40.0–50.0 | 2.26 | 1.28 | 0.10 | 1.01 | 1.05 | 1.11 | 1.14 | 1.19 | 1.28 | 1.36 | 1.40 | 1.43 | 1.47 | 1.49 |
| 50.0–60.0 | 2.34 | 1.27 | 0.10 | 1.00 | 1.04 | 1.10 | 1.13 | 1.18 | 1.27 | 1.35 | 1.39 | 1.42 | 1.46 | 1.48 |
| 60.0–70.0 | 2.42 | 1.26 | 0.10 | 0.99 | 1.03 | 1.09 | 1.12 | 1.17 | 1.26 | 1.34 | 1.38 | 1.41 | 1.44 | 1.47 |
| 70.0–80.0 | 2.49 | 1.25 | 0.10 | 0.98 | 1.02 | 1.08 | 1.11 | 1.16 | 1.25 | 1.33 | 1.37 | 1.39 | 1.43 | 1.45 |
| **Females** | | | | | | | | | | | | | | |
| 5.0–5.9 | −2.00 | 0.69 | 0.09 | 0.60 | 0.61 | 0.62 | 0.63 | 0.65 | 0.69 | 0.74 | 0.77 | 0.79 | 0.83 | 0.86 |
| 6.0–6.9 | −1.59 | 0.73 | 0.09 | 0.62 | 0.63 | 0.65 | 0.66 | 0.68 | 0.73 | 0.77 | 0.81 | 0.83 | 0.86 | 0.89 |
| 7.0–7.9 | −1.18 | 0.76 | 0.09 | 0.65 | 0.66 | 0.68 | 0.69 | 0.72 | 0.76 | 0.81 | 0.84 | 0.86 | 0.90 | 0.92 |
| 8.0–8.9 | −0.78 | 0.80 | 0.09 | 0.68 | 0.69 | 0.71 | 0.73 | 0.75 | 0.80 | 0.85 | 0.88 | 0.90 | 0.94 | 0.96 |
| 9.0–9.9 | −0.39 | 0.84 | 0.09 | 0.71 | 0.72 | 0.74 | 0.76 | 0.79 | 0.84 | 0.89 | 0.92 | 0.94 | 0.98 | 1.00 |
| 10.0–10.9 | −0.04 | 0.88 | 0.09 | 0.74 | 0.76 | 0.78 | 0.80 | 0.83 | 0.88 | 0.94 | 0.97 | 0.99 | 1.03 | 1.05 |
| 11.0–11.9 | 0.28 | 0.93 | 0.09 | 0.78 | 0.79 | 0.82 | 0.84 | 0.87 | 0.93 | 0.99 | 1.02 | 1.04 | 1.07 | 1.10 |
| 12.0–12.9 | 0.54 | 0.97 | 0.09 | 0.81 | 0.83 | 0.86 | 0.88 | 0.91 | 0.97 | 1.03 | 1.07 | 1.09 | 1.12 | 1.15 |
| 13.0–13.9 | 0.74 | 1.01 | 0.09 | 0.84 | 0.86 | 0.90 | 0.92 | 0.95 | 1.01 | 1.07 | 1.11 | 1.13 | 1.17 | 1.19 |
| 14.0–14.9 | 0.88 | 1.05 | 0.09 | 0.87 | 0.89 | 0.93 | 0.95 | 0.98 | 1.05 | 1.11 | 1.14 | 1.17 | 1.20 | 1.22 |
| 15.0–15.9 | 0.96 | 1.07 | 0.09 | 0.90 | 0.92 | 0.95 | 0.98 | 1.01 | 1.07 | 1.14 | 1.17 | 1.20 | 1.23 | 1.25 |
| 16.0–16.9 | 0.98 | 1.10 | 0.09 | 0.91 | 0.94 | 0.97 | 1.00 | 1.03 | 1.10 | 1.16 | 1.20 | 1.22 | 1.26 | 1.28 |
| 17.0–17.9 | 0.96 | 1.11 | 0.09 | 0.93 | 0.95 | 0.99 | 1.01 | 1.05 | 1.11 | 1.18 | 1.21 | 1.24 | 1.27 | 1.30 |
| 18.0–18.9 | 0.91 | 1.12 | 0.09 | 0.94 | 0.96 | 1.00 | 1.02 | 1.06 | 1.12 | 1.19 | 1.23 | 1.25 | 1.29 | 1.31 |
| 19.0–19.9 | 0.84 | 1.13 | 0.09 | 0.94 | 0.96 | 1.00 | 1.02 | 1.06 | 1.13 | 1.20 | 1.24 | 1.26 | 1.30 | 1.32 |
| 20.0–30.0 | 0.78 | 1.13 | 0.09 | 0.93 | 0.96 | 0.99 | 1.02 | 1.06 | 1.13 | 1.20 | 1.24 | 1.26 | 1.30 | 1.33 |
| 30.0–40.0 | 0.71 | 1.12 | 0.10 | 0.92 | 0.94 | 0.98 | 1.01 | 1.04 | 1.12 | 1.19 | 1.23 | 1.25 | 1.30 | 1.32 |
| 40.0–50.0 | 0.66 | 1.09 | 0.10 | 0.90 | 0.92 | 0.96 | 0.98 | 1.02 | 1.09 | 1.17 | 1.21 | 1.24 | 1.28 | 1.31 |
| 50.0–60.0 | 0.60 | 1.07 | 0.10 | 0.87 | 0.89 | 0.93 | 0.95 | 0.99 | 1.07 | 1.14 | 1.18 | 1.21 | 1.25 | 1.28 |
| 60.0–70.0 | 0.54 | 1.03 | 0.11 | 0.83 | 0.86 | 0.89 | 0.92 | 0.96 | 1.03 | 1.11 | 1.15 | 1.18 | 1.22 | 1.25 |
| 70.0–80.0 | 0.47 | 1.00 | 0.11 | 0.80 | 0.82 | 0.86 | 0.88 | 0.92 | 1.00 | 1.07 | 1.11 | 1.14 | 1.19 | 1.22 |

**Note:**

L, Lambda; M, median; S, Sigma.

**Table 3 Total body (g) bone mineral content from age 5.0 to 80 years old of subjects from a province of Chile.**

| Age | BMC | | | | | | | | | | | | | |
|---|---|---|---|---|---|---|---|---|---|---|---|---|---|---|
| | L | M | S | P3 | P5 | P10 | P15 | P25 | P50 | P75 | P85 | P90 | P95 | P97 |
| **Males** | | | | | | | | | | | | | | |
| 5.0–5.9 | −0.16 | 0.75 | 0.02 | 0.56 | 0.57 | 0.60 | 0.63 | 0.66 | 0.75 | 0.87 | 0.96 | 1.03 | 1.18 | 1.30 |
| 6.0–6.9 | −0.13 | 0.86 | 0.02 | 0.63 | 0.65 | 0.69 | 0.71 | 0.76 | 0.86 | 0.99 | 1.08 | 1.16 | 1.29 | 1.41 |
| 7.0–7.9 | −0.10 | 0.97 | 0.02 | 0.71 | 0.73 | 0.78 | 0.81 | 0.86 | 0.97 | 1.12 | 1.21 | 1.29 | 1.42 | 1.52 |
| 8.0–8.9 | −0.07 | 1.09 | 0.02 | 0.79 | 0.82 | 0.87 | 0.91 | 0.97 | 1.09 | 1.25 | 1.35 | 1.43 | 1.56 | 1.65 |
| 9.0–9.9 | −0.04 | 1.24 | 0.02 | 0.88 | 0.92 | 0.98 | 1.02 | 1.09 | 1.24 | 1.41 | 1.51 | 1.59 | 1.72 | 1.81 |
| 10.0–10.9 | −0.01 | 1.40 | 0.02 | 0.99 | 1.03 | 1.10 | 1.16 | 1.24 | 1.40 | 1.59 | 1.70 | 1.78 | 1.91 | 2.00 |
| 11.0–11.9 | 0.02 | 1.59 | 0.02 | 1.11 | 1.16 | 1.25 | 1.31 | 1.40 | 1.59 | 1.80 | 1.91 | 2.00 | 2.13 | 2.22 |
| 12.0–12.9 | 0.05 | 1.80 | 0.02 | 1.25 | 1.31 | 1.41 | 1.48 | 1.59 | 1.80 | 2.03 | 2.15 | 2.24 | 2.37 | 2.46 |
| 13.0–13.9 | 0.07 | 2.03 | 0.02 | 1.39 | 1.47 | 1.59 | 1.67 | 1.79 | 2.03 | 2.27 | 2.40 | 2.49 | 2.63 | 2.72 |
| 14.0–14.9 | 0.10 | 2.25 | 0.02 | 1.54 | 1.63 | 1.76 | 1.86 | 1.99 | 2.25 | 2.51 | 2.65 | 2.74 | 2.88 | 2.98 |
| 15.0–15.9 | 0.11 | 2.46 | 0.02 | 1.68 | 1.78 | 1.93 | 2.03 | 2.18 | 2.46 | 2.73 | 2.88 | 2.97 | 3.12 | 3.21 |
| 16.0–16.9 | 0.12 | 2.64 | 0.02 | 1.81 | 1.92 | 2.08 | 2.19 | 2.35 | 2.64 | 2.92 | 3.07 | 3.17 | 3.31 | 3.40 |
| 17.0–17.9 | 0.13 | 2.78 | 0.02 | 1.91 | 2.03 | 2.20 | 2.32 | 2.48 | 2.78 | 3.06 | 3.21 | 3.31 | 3.46 | 3.55 |
| 18.0–18.9 | 0.14 | 2.88 | 0.02 | 2.00 | 2.12 | 2.30 | 2.41 | 2.58 | 2.88 | 3.17 | 3.32 | 3.42 | 3.57 | 3.66 |
| 19.0–19.9 | 0.15 | 2.96 | 0.01 | 2.06 | 2.18 | 2.37 | 2.48 | 2.65 | 2.96 | 3.25 | 3.40 | 3.50 | 3.64 | 3.74 |
| 20.0–30.0 | 0.15 | 3.00 | 0.01 | 2.11 | 2.23 | 2.41 | 2.53 | 2.70 | 3.00 | 3.29 | 3.44 | 3.54 | 3.68 | 3.77 |
| 30.0–40.0 | 0.16 | 3.02 | 0.01 | 2.13 | 2.25 | 2.43 | 2.55 | 2.72 | 3.02 | 3.30 | 3.45 | 3.54 | 3.69 | 3.78 |
| 40.0–50.0 | 0.16 | 3.01 | 0.01 | 2.14 | 2.26 | 2.44 | 2.55 | 2.72 | 3.01 | 3.29 | 3.43 | 3.52 | 3.66 | 3.75 |
| 50.0–60.0 | 0.16 | 2.99 | 0.01 | 2.13 | 2.25 | 2.43 | 2.54 | 2.70 | 2.99 | 3.25 | 3.39 | 3.48 | 3.62 | 3.70 |
| 60.0–70.0 | 0.17 | 2.95 | 0.01 | 2.12 | 2.24 | 2.41 | 2.52 | 2.68 | 2.95 | 3.21 | 3.35 | 3.44 | 3.56 | 3.64 |
| 70.0–80.0 | 0.17 | 2.92 | 0.01 | 2.11 | 2.22 | 2.39 | 2.50 | 2.65 | 2.92 | 3.17 | 3.30 | 3.38 | 3.51 | 3.58 |
| **Females** | | | | | | | | | | | | | | |
| 5.0–5.9 | −0.12 | 0.72 | 0.02 | 0.52 | 0.54 | 0.57 | 0.59 | 0.63 | 0.72 | 0.84 | 0.93 | 1.00 | 1.13 | 1.24 |
| 6.0–6.9 | −0.09 | 0.84 | 0.02 | 0.60 | 0.62 | 0.66 | 0.69 | 0.74 | 0.84 | 0.97 | 1.06 | 1.13 | 1.26 | 1.35 |
| 7.0–7.9 | −0.06 | 0.96 | 0.02 | 0.69 | 0.72 | 0.76 | 0.79 | 0.85 | 0.96 | 1.11 | 1.20 | 1.27 | 1.39 | 1.48 |
| 8.0–8.9 | −0.03 | 1.10 | 0.02 | 0.78 | 0.81 | 0.87 | 0.91 | 0.97 | 1.10 | 1.25 | 1.35 | 1.42 | 1.54 | 1.62 |
| 9.0–9.9 | −0.01 | 1.25 | 0.02 | 0.88 | 0.92 | 0.98 | 1.03 | 1.10 | 1.25 | 1.41 | 1.51 | 1.58 | 1.69 | 1.77 |
| 10.0–10.9 | 0.02 | 1.40 | 0.02 | 0.99 | 1.04 | 1.11 | 1.16 | 1.24 | 1.40 | 1.58 | 1.68 | 1.75 | 1.87 | 1.94 |
| 11.0–11.9 | 0.04 | 1.56 | 0.02 | 1.11 | 1.16 | 1.24 | 1.30 | 1.39 | 1.56 | 1.75 | 1.85 | 1.93 | 2.04 | 2.11 |
| 12.0–12.9 | 0.06 | 1.71 | 0.02 | 1.22 | 1.28 | 1.37 | 1.43 | 1.53 | 1.71 | 1.91 | 2.02 | 2.09 | 2.20 | 2.28 |
| 13.0–13.9 | 0.08 | 1.85 | 0.02 | 1.32 | 1.38 | 1.48 | 1.55 | 1.65 | 1.85 | 2.05 | 2.15 | 2.23 | 2.34 | 2.41 |
| 14.0–14.9 | 0.09 | 1.95 | 0.02 | 1.41 | 1.48 | 1.58 | 1.65 | 1.76 | 1.95 | 2.16 | 2.26 | 2.34 | 2.45 | 2.52 |
| 15.0–15.9 | 0.09 | 2.04 | 0.01 | 1.49 | 1.55 | 1.66 | 1.73 | 1.84 | 2.04 | 2.24 | 2.35 | 2.42 | 2.53 | 2.60 |
| 16.0–16.9 | 0.10 | 2.10 | 0.01 | 1.55 | 1.62 | 1.72 | 1.80 | 1.90 | 2.10 | 2.31 | 2.41 | 2.49 | 2.60 | 2.67 |
| 17.0–17.9 | 0.10 | 2.15 | 0.01 | 1.60 | 1.67 | 1.77 | 1.85 | 1.95 | 2.15 | 2.36 | 2.46 | 2.54 | 2.64 | 2.72 |
| 18.0–18.9 | 0.10 | 2.19 | 0.01 | 1.63 | 1.70 | 1.81 | 1.88 | 1.99 | 2.19 | 2.39 | 2.50 | 2.57 | 2.68 | 2.75 |
| 19.0–19.9 | 0.10 | 2.21 | 0.01 | 1.65 | 1.72 | 1.83 | 1.90 | 2.01 | 2.21 | 2.42 | 2.52 | 2.60 | 2.71 | 2.78 |
| 20.0–30.0 | 0.10 | 2.22 | 0.01 | 1.64 | 1.72 | 1.83 | 1.90 | 2.01 | 2.22 | 2.42 | 2.53 | 2.60 | 2.71 | 2.78 |
| 30.0–40.0 | 0.11 | 2.20 | 0.01 | 1.62 | 1.69 | 1.80 | 1.88 | 1.99 | 2.20 | 2.40 | 2.51 | 2.59 | 2.69 | 2.76 |
| 40.0–50.0 | 0.12 | 2.16 | 0.01 | 1.57 | 1.64 | 1.76 | 1.84 | 1.95 | 2.16 | 2.36 | 2.47 | 2.54 | 2.65 | 2.72 |
| 50.0–60.0 | 0.13 | 2.10 | 0.01 | 1.50 | 1.58 | 1.70 | 1.78 | 1.89 | 2.10 | 2.30 | 2.41 | 2.48 | 2.59 | 2.65 |
| 60.0–70.0 | 0.13 | 2.02 | 0.02 | 1.42 | 1.50 | 1.62 | 1.70 | 1.82 | 2.02 | 2.23 | 2.33 | 2.40 | 2.50 | 2.57 |
| 70.0–80.0 | 0.14 | 1.94 | 0.02 | 1.34 | 1.42 | 1.54 | 1.62 | 1.74 | 1.94 | 2.14 | 2.25 | 2.31 | 2.42 | 2.48 |

**Note:**
BMC, bone mineral content; L, Lambdal; M, median; S, Sigma.

The PBM determined from BMD was estimated for males as 1.45 ± 0.18 g/cm$^2$ and for females as 1.22 ± 0.13 g/cm$^2$. These mean values were greater than those reported for other studies (*Looker et al., 2012*; *Tamayo et al., 2009*; *Chen et al., 2020*). However, with regard to the age that the PBM was observed, it needs to be highlighted that for the regional population of the Maule (Chile), PBM appeared at approximately 28 years old for females and age 30 for males. This was similar to those reported for the American references NHANES (*Looker et al., 2012*), Mexican populations (*Tamayo et al., 2009*), and those from Singapore (*Chen et al., 2020*). Other research studies using DXA a have reported an earlier age attainment of PBM, being 18 to 20 years, respectively (*Nakavachara et al., 2014*; *Yi et al., 2014*; *van der Sluis et al., 2002*).

In general, to date, no consensus has been reached on at what age PBM is reached in both sexes, and it currently remains uncertain (*Kralick & Zemel, 2020*; *Lu et al., 2016*). However, the high mean values observed in this study could contribute information for the future (*Mølgaard et al., 1997*), as they are important for guidance on preventive approaches to bone loss and osteoporosis.

Optimizing bone mass acquisition by 10%, could benefit older adults (*Cummings et al., 1993*) and substantially reduce the risk of fractures and increased healthcare costs (*Gitlin & Fuentes, 2012*).

Recent studies have suggested optimizing the acquisition of bone mass during the growth and development stages, such as in the preventing of current or future fractures as bone mass, density, and structural strength are associated with fractures in children and adolescents (*Kalkwarf, Laor & Bean, 2011*). In addition, some modifiable factors, such as diet, physical activity, hormonal regulation, and smoking, can dramatically contribute to the decrease in peak bone mass (*Krall & Dawson-Hughes, 1993*; *Abrahamsen et al., 2014*). Therefore, improving skeletal health throughout life is a multifactorial endeavor, with no single factor driving bone mass loss and/or gain.

Based on previous findings, this current study proposed developing percentiles for BMD and BMC for the regional population of |Maule (Chile). In this context, several studies have highlighted that bone health references need to be specific for each sex and ethnicity (*Zhang et al., 2014*; *Ho-Pham et al., 2011*; *Heidari et al., 2016*). This may help with correctly diagnosing and treating suspected cases of low levels of bone mass. It may also help avoid an inadequate or excessive diagnosis of osteopenia and/or osteoporosis (*Nakavachara et al., 2014*; *Gilsanz et al., 1998*), respectively.

In general, the percentiles proposed in this study, like the cubic polynomial regression, showed three phases in bone health (BMD and BMC) from age 5 until 80 years old in both sexes. The first phase of accelerated growth is highlighted from age 5 to age 30. This is followed by a second phase of stability lasting until the beginning of age 40, and a third phase characterized by a gradual decrease in bone health until 80 years old.

The pattern observed in this study is consistent with other research suggesting the importance of monitoring BMD and BMC throughout life. This, especially during early growth and development, which is when bone mass accumulates rapidly (*Chen et al., 2020*; *Ho-Pham et al., 2011*; *Heidari et al., 2016*).

Population references based on DXA for BMD and BMC are scarce, not only for Chile but also for South America in general. Thus, the use of references could be useful to assess bone growth abnormalities in the prevention and follow-up of osteoporosis (*Lu et al., 2016*).

Therefore, the reference data proposed in this study may be needed for clinical care, treatment, and prevention for future bone health. In addition, it could be used for early detection of osteoporosis, facilitating treatment before a fracture occurs due to bone fragility (*Kling, Clarke & Sandhu, 2014*).

Our research has a number of strong points. For example, it is the first cross-sectional study of skeletal health conducted in Chile and South America using DXA. In our research, the sample size consisted of 5,416 individuals. This made it possible to analyze the data obtained from age 5 to 80 years old. In addition, DXA was used as the gold standard for evaluating the BMD and BMC. These results obtained in this research can be used for Chilean populations. The database from this study could be used as a valuable reference for health professionals and researchers since the data could be used as a base line for future comparisons of secular trends and comparisons with international studies. In addition, the results presented here can be used on a daily basis to determine bone mineral measurements through the following link: http://reidebihu.net/bmd_bmc_5_80ch.php. This website is for professionals working with DXA, where personal data and BMD and BMC values are entered to be compared with the reference. The website allows to print the results.

In terms of limitations, the ideal would be to study bone mass throughout life longitudinally. However, the cross-sectional design used did not allow us to identify causal relationships; moreover, it was not possible to evaluate some variables, such as dietary habits, levels of physical activity and calcium and vitamin D intake. It should also be noted that the comparisons made with the US study may differ in lifestyles and in the type of equipment used.

As a result, the findings obtained from this research need to be analyzed with caution. Future studies need to overcome these limitations identified, and the sample needs to be broadened, and include references from birth to 4 years old.

## CONCLUSION

In conclusion, this study identified differences in bone mineral measurements with reference data from the United States. Furthermore, the study identified an accelerated growth phase of bone health up to 28–30 years old (reaching maximum bone mass at age 28 in females and 30 years old in males), followed by a second phase of stability until age 40 to later diminish considerably in the third phase until age 80. In order to take advantage of the database, substantial reference values were proposed for evaluating bone health and diagnosing fragility and osteoporosis for individuals and regional groups and populations of the Maule in Chile.

### Funding

This work was supported by the Fondecyt ANID 1141295. The funders had no role in study design, data collection and analysis, decision to publish, or preparation of the manuscript.

### Grant Disclosures

The following grant information was disclosed by the authors:
The Fondecyt ANID 1141295.

### Competing Interests

The authors declare that they have no competing interests.

### Author Contributions

- Marco Cossio-Bolanos conceived and designed the experiments, analyzed the data, prepared figures and/or tables, authored or reviewed drafts of the paper, and approved the final draft.
- Rubén Vidal-Espinoza performed the experiments, analyzed the data, authored or reviewed drafts of the paper, and approved the final draft.
- Jose Fuentes-Lopez analyzed the data, authored or reviewed drafts of the paper, and approved the final draft.
- Luis Felipe Castelli Correia de Campos performed the experiments, prepared figures and/or tables, and approved the final draft.
- Cynthia Lee Andruske performed the experiments, authored or reviewed drafts of the paper, and approved the final draft.
- Camilo Urra-Albornoz performed the experiments, prepared figures and/or tables, and approved the final draft.
- Fernando Alvear Vasquez performed the experiments, prepared figures and/or tables, and approved the final draft.
- Rossana Gomez-Campos conceived and designed the experiments, analyzed the data, prepared figures and/or tables, and approved the final draft.

### Human Ethics

The following information was supplied relating to ethical approvals (*i.e.*, approving body and any reference numbers):

The Autonomous University approved the study.

### Data Availability

The raw data is available in the Supplemental File.

## Supplemental Information

Supplemental information for this article can be found online at http://dx.doi.org/10.7717/peerj.13092#supplemental-information.

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
