# Peer review of "Reference values for bone density and bone mineral content from 5 to 80 years old in a province of Chile"

_PeerJ, doi:10.7717/peerj.13092_

## Round 0.1 · original submission · Major Revisions

The reviewers have an overall positive impression of your work and provided detailed comments and suggested changes.

Reviewer 1 ·

Basic reporting

Much of my review includes editorial revisions (i.e., statements that can be streamlined, unclear, grammatically incorrect phrasing etc.) as well as some more major issues in terms of discussion of findings. See below for details:


This manuscript “Reference values for bone density and bone mineral content from 50 to 80 years in the province of Chile” offers important and much-needed insight onto skeletal health across global populations, for which there are limited data available. Reference values that are regionally specific are crucial for guiding health policy and practice that target the population in question. The sample size in this manuscript is massive, across many critical age cohorts, and though this is a cross-sectional study, it sheds light on patterns of bone accrual and loss across the life course. Ultimately, this manuscript can make an important contribution to skeletal health literature, in addition to the website they created which may serve as a useful reference for international scholars.


That being said, there are major revisions must be made before being considered for publication. Most of the important revisions that I suggest are particular to the framing, results and discussion. This includes but is not limited to: 1) Framing the importance of this study more directly (i.e., emphasizing why local population-specific references are needed); 2) Performing a statistical analysis that directly compares US and Chilean mean values for BMC and BMD; 3) Clearly defining BMC and BMD in the manuscript; and 4) Streamlining the discussion to more clearly reflect the results of this study. Specifics as well as other suggested revisions are highlighted below:

INTRODUCTION:
• First paragraph: This opening paragraph is framed in a way that suggests children and adolescents are going to be the main focus of this paper. Yet this paper doesn’t emphasize subadults any more than adults. I think a more compelling way to open the paper is to note the importance of regionally/population specific reference values and the dearth of this information for Chilean and South American populations more broadly.
o The authors may also highlight that there are relatively few studies that examine skeletal health, using the same methodology, across subadults and adults. I think this is one of the most valuable contributions of this current manuscript.

• Line 56-57: The terms bone mineral content and bone mineral density need to be defined early on in this paper in the same place. Here BMC is mentioned but not BMD; why not?
o Also the sentence beginning with “This stage…” is inaccurate in that growth is a process rather than a stage. Please clarify.

• Line 60: It’s better to be specific and note that accumulation of bone density and content establishes the foundation for skeletal health throughout the lifespan. Simply stating that these are “keys for skeletal health” is vague.

• Line 62-65: Consider revising the wording here. Suggestion: “Furthermore, research indicates that more than half of the peak bone mass…”
o Also, I understand the term “maximum bone mass” but it’s more commonly referred to as “peak bone mass” (PBM). Moreover, the abbreviation for maximum bone mass (MBM) is introduced in the discussion in line 265. The abbreviation should be introduced first here, in the beginning of the manuscript.
o The last sentence beginning with “In addition,…” is confusing and it may be worthwhile to revise without the 7 to 11%. Suggestion: “The remainder of adult bone mass is acquired after linear growth ceases”

• Line 66-70: Consider revising the wording here as it can be simplified for clarity. Suggestion “The specific age at peak bone mass achievement is unclear as data are inconsistent”.
o The statement that other studies “argue that [peak bone mass] is reached in an individual’s third decade or beyond” is not a contrasting result from the previous sentence where the authors note peak bone mass is achieved between 25-30 years old. 25-30 years old is the third decade of life. So consider revising this sentence (line 69). Suggestion: “However, other studies conclude that peak bone mass may be reached beyond the third decade of life”.

• Line 71-74: This seems out of place and should be moved to the beginning of the paper when peak/maximum bone mass is first mentioned (right after line 65).

• Line 75: delete “the” before maximum bone mass; also, instead of “independent of the age where it presents” which is a confusing statement, consider revising to “regardless of age of achievement”

• Line 77-78. Instead of “from one individual to another”, consider revising to “between individuals” to reduce wordiness. More importantly, the authors mention “distinct patterns according to age and sex” of bone loss. An example should be presented here such as (for i.e., female loss is accelerated compared to male loss) or something similar. I realize the authors can’t present all the patterns here but one example, at least, is warranted.

• Line 84-87: Instead of “threats to bone mineral measurements”, consider “disruptions to skeletal health (BMD, BMC). Again, these need to be defined earlier in the paper. Furthermore, if bone area is not directly studied as a variable in this paper, it would be good to remove it from this paragraph altogether.

o The sentence beginning with “The importance of assessing these variables during growth…” is not clear and is excessively wordy (which may be why it’s unclear). Consider revising to more simply say “Examining these variables across the life course may be critical for understanding and attenuating fracture risk in advanced ages”.

• Line 88-92: I don’t think this paragraph is needed. It doesn’t add anything to the manuscript. I think the important thing to note is that there are these international references available but local, regionally-specific or population-specific references are more valuable for guiding health policy and practice.

• Line 95-97: It’s unclear what the authors mean in this sentence beginning with “However, it seems that…” In particular, what is meant by “a higher proportion of reference values for pediatric populations”. Again, I think it’s necessary to emphasize that local references are ideal because of the heterogeneity of global bodies (for example US NHANES data isn’t ideal for use globally). And moreover, the data in this current manuscript sheds light onto skeletal health across the lifespan.

• Line 100-101. A bit more expansion is needed here – in particular, a description of the demographic transition here. The assumption is that Chile is characterized by an expanding aging population and a shrinking subadult population but this needs to be stated.

METHODS
• Line 115: Is 41,200 the number of people in the province or the study population from which a randomized sample was pulled? Figure 2 states that this number is the “total population of the province” but in this line, it’s not stated as such. Please clarify.

• Line 117-118: It would be helpful to give age ranges for the categories of participants. For example, what age range are participants considered “children”, “adolescents” etc. And moreover, what does “youth” include and how is that different from “children and adolescents”? This term youth is not specific enough and likely redundant.

• Lines 128-132: I don’t think it’s necessary to include Chile’s HDI here. If it must be included, provide a reference for understanding the index of 0.847. Is this high? Is this low?

o The education system is described as being divided into four phases yet three are listed, unless a comma is missing: early childhood, primary media (is the comma missing here?), and higher education? Although ultimately, I don’t think this information is entirely necessary in the manuscript.

• Line 135-136. How was nationality determined? By passport? Did participants self identify? Were all participants born in Chile? Please elaborate briefly on this.

• Line 138-139: Delete the zero before the number of children and adults excluded from the study (i.e., 4 children, 6 adults).

• Line 166: Provide ages for what qualifies as pediatric populations (i.e, < 18 or < 21 years old).

• Line 175: What is a test-re-test? I think these two sentences can be combined and rephrased as: “To ensure quality control for the variables collected, 10% of the total number of participants were evaluated twice for both anthropometry and DXA scans.”

• Line 187: This should be reworded as “Differences between sexes” and not genders.

• Line 193: Regarding R2, the 2 should be a superscript: R2


RESULTS
• Line 208-209: The authors can simply state from 13 to 80 years old (and not 13 to 70-80).
o Also, consider revising the wording here to simplify. Suggestion “…females had lower BMC compared to males. No significant differences in BMI between sexes were identified.”

• Line 212-214. The comparison between Chilean females and US females in this sentence is not clearly described. A few words are missing (i.e., the mean values for BMD were relatively similar compared to US females?)
o Also, there should be a statistical test included in the methods that compares the US and Chilean means across the life course.

• Line 221-223. I just want to confirm that the age at peak bone mass achievement is the same age for both BMD and BMC in males and females. So males achieve peak bone mineral density AND content at age 30 and females at age 28 for both variables?

• Line 225-227: Consider revising these sentences for clarity: Suggestion “…followed by a plateau with no significant gains or losses until 40 years old. After age 40, BMD and BMC undergo declines that continue through to advanced age.”


• Line 229-232: These sentences can be streamlined. Suggestion: “The values for BMD in both sexes were similar from age 5 to 13 after which males have a higher median compared to females. This difference persists until advanced age. However, males have higher BMC values compared to females across all ages with the greatest difference manifesting at 10 years old and persisting until 80 years old.”

DISCUSSION:
• Line 237-239. Consider revising for clarity and succinctness. Suggestion: “The results from this study demonstrate that Chilean males from Maule have marginally higher BMC and BMD values compared to US populations, based on NHANES data while Chilean females display similar values to that of US females.”
o In addition to revising wording for clarity, it is important to repeat that no statistical test of these differences were presented in the results and this should be remedied.

• Line 242-249: The entire first paragraph of the discussion compares BMC and BMD values from this study to other studies with no mention of a) the methodology of these studies (i.e., were these all DXA based studies?) And b) no statistical analysis performed to confirm these comparisons. While I don’t think statistical comparisons are needed for all of the references mentioned, at the minimum, some statistical comparison between this study population and the US NHANES is warranted.

• Line 250- 258. These two sentences can be combined and streamlined. Suggestion: “These findings are consistent with other research that have reported population differences in BMD and BMC across the life course.”

o This sentence can be followed up with this suggested revision “Differences in skeletal health values across diverse geographical and ethnic populations may be due to various factors including diet, nutritional status….etc.”
o Also, do the authors mean “timing of” sexual maturation? What do they mean by “sexual maturation” as a factor related to skeletal health?

• Line 265: As mentioned earlier, MBM or preferably PBM should be introduced as an abbreviation in the beginning of the manuscript.

• Line 265: Here, the authors state that MBM is determined from maximum BMD. However, this needs to be stated earlier on in the manuscript, in the methodology, or in the introduction. Currently MBM is defined in lines 71-74 as a broad term encompassing mass, density, micro-architecture etc. but then gets distilled as maximum BMD in this paper. So a justification needs to be made for this, however, brief, that here, maximum bone mass is really maximum bone density.

• Line 270-272. Instead of just stating that MBM in this study was similar to US, Mexican, and Singaporean populations, providing those ages identified in these studies would be helpful (as they do in the next sentence).

o Instead of beginning the sentence with “Although, evidently, other research studies have” consider revising to: “Other research studies have reported younger age at peak bone mass achievement as young as 18-20 years old (Nakavachara et al., 2014, Yi et al., 2014, van Der Sluis et al., 2002)”
o Also, please clarify if the above studies used DXA scans.

• Lines 275-278: Can the authors state why a specific age at MBM achievement is important? Line 277-278 begins to highlight the importance but falls short. Why are these values important for informing preventative approaches for bone loss and osteoporosis? This needs to be made more clear throughout.

• Line 279-282: The abbreviation PMO is introduced without any explanation or previous use in the manuscript.

o This sentence is redundant and needs to be streamlined altogether.

• Line 283-287: These sentences are very confusing and come across as rambling. Please clarify – eliminating some words and stating simply that finding ways to maximize bone mass by 10% could benefit older adults and could reduce the risk of fractures and heightened health care costs substantially.

• Line 288: Remove the words “In this context” as it doesn’t make sense following the last paragraph. Simply beginning with “Recent studies have suggested…” would suffice.

• Line 294: The statement “Therefore, it is necessary to improve lifestyles” is a vague, broad, and not really helpful conclusion. I would delete this altogether as specific modifiable factors have already been presented. Also, it would be useful to emphasize here that ameliorating bone health across the life course is a multi-factorial endeavor, with no single factor driving bone loss and gain.

• Line 296-299: This paragraph begins by discussion the percentiles determined in the study and then switches focus to environmental factors and ends with “these need to be permanently monitored”. None of these statements make sense here – more consideration of the percentiles presented is warranted (for some reason lines 305-309 elaborate on the percentiles but this section can be moved up here). Lines 297-299 can be deleted altogether as they don’t contribute anything new to the discussion.

• Line 303-304: Osteopenia should be defined briefly here (i.e., low bone mass) as well as osteoporosis.

• Line 310-314: Again, this section is rambling and needs some focus. Consider revising some of the wording here. Suggestion: “The pattern observed in this study is consistent with other research suggesting the importance in monitoring BMD and BMC across the life course, particularly during early growth and development when bone mass is rapidly being accrued”.

• Line 315-318: This should be reframed as “DXA-based population references for BMD and BMC are scare not only for Chile, but South America in general” as I know of a few QUS based references for South American populations.

• Line 323-325: Please qualify the statement “This is the first cross-sectional study to be carried out in Chile as well as South America”. First, do the authors mean “DXA-based skeletal health study”? Keep in mind that I am aware of a number of non-DXA based bone health studies from South America, as well as a few DXA studies. So, this statement needs to be expanded upon.

• Line 326: Why use the words “It is possible that these results may be used for Chilean populations”. If I understand this correctly the authors mean that these data can be useful for Chilean populations more broadly – I’d eliminate the words “it is possible” in this statement because clearly local references are more applicable than international references.

• Line 330-332: I’m not clear about what the website is presenting based on the manuscript description here. Is this website for participants to see their results? Or for people interested in seeing where their values fall in relation to the study participants? Please clarify the utility of this website (which has value!) in the manuscript.

• Line 333 – 339: This is an awkward beginning to a sentence. The authors simply need to state that ideally, bone mass across the life course would be studied longitudinally.

o Include the sample size for males between 30-80 vs. females if it is going to be mentioned as a limitation to this study.
o The sentence beginning with “As well, it was not possible to evaluate some variables…” should be moved up where right after “…identify causal relationships”.

FIGURES
• In figure 2, the total number of excluded participants is indicated as 104 although the numbers added up equals 103.

TABLES
• The tables have a lot of data in them and are visually hard to track – however one thing that can be easily addressed is emphasizing MALE and FEMALE in the table by bolding the words or finding some other way to emphasize these sections.

ABSTRACT
• In the first sentence of the background, “fractures” risk should be “fracture” risk.
• In the results and discussion section, instead of “Discrepancies” consider revising to: “Differences between Chilean and US male mean values were identified for BMD and BMC.” Or something similar to streamline the sentence.

Experimental design

As indicated in my response to 1) most of the important revisions that I suggest are particular to the framing, results and discussion. This includes but is not limited to: 1) Framing the importance of this study more directly (i.e., emphasizing why local population-specific references are needed); 2) Performing a statistical analysis that directly compares US and Chilean mean values for BMC and BMD; 3) Clearly defining BMC and BMD in the manuscript; and 4) Streamlining the discussion to more clearly reflect the results of this study.

Validity of the findings

The findings are sound but a more clear, streamlined discussion of the results is warranted. Additionally, a statistical analysis of the US and Chilean mean values for BMC and BMD should be included rather than simply stating "they look similar".

Additional comments

see other sections for specific details.

Reviewer 2 ·

Basic reporting

The aim of the study is clear
The design is adequate
Population is large enough and well stratified

Experimental design

It is well designed and population stratified

Validity of the findings

Validity of the findings are right
I have only two observations
1 . Individuals between 5 and 25 years of age should be correlated with Tanner score of pubertal development rather than with chronological age
2 BMD and peak bone mass should be correlated with calcium intake and nutritional data .

Additional comments

It is a good study
My observations are
1 BMD should be correlated with Tanner score of pubertal development in individuals between 5 and 25 years of age
2 BMD should be correlated with calcium intake and nutritional data

---

## Round 0.2 · accepted · Accept

Thank you for your revised submission in which you have addressed all the concerns of the reviewers.